# High resolution DNA barcode library for European butterflies reveals continental patterns of mitochondrial genetic diversity

Vlad Dincă [1,2,10 ✉], Leonardo Dapporto [3,10], Panu Somervuo[4], Raluca Vodă[5], Sylvain Cuvelier[6], Martin Gascoigne-Pees[7], Peter Huemer[8], Marko Mutanen[1], Paul D. N. Hebert[9] & Roger Vila [2]

The study of global biodiversity will greatly benefit from access to comprehensive DNA barcode libraries at continental scale, but such datasets are still very rare. Here, we assemble the first high-resolution reference library for European butterflies that provides 97% taxon coverage (459 species) and 22,306 *COI* sequences. We estimate that we captured 62% of the total haplotype diversity and show that most species possess a few very common haplotypes and many rare ones. Specimens in the dataset have an average 95.3% probability of being correctly identified. Mitochondrial diversity displayed elevated haplotype richness in southern European refugia, establishing the generality of this key biogeographic pattern for an entire taxonomic group. Fifteen percent of the species are involved in barcode sharing, but two thirds of these cases may reflect the need for further taxonomic research. This dataset provides a unique resource for conservation and for studying evolutionary processes, cryptic species, phylogeography, and ecology.

[1] Ecology and Genetics Research Unit, PO Box 3000, University of Oulu, 90014 Oulu, Finland. [2] Institut de Biologia Evolutiva (CSIC-UPF), 03008 Barcelona, Spain. [3] ZEN lab, Dipartimento di Biologia, University of Florence, 50019 Sesto Fiorentino, Italy. [4] Organismal and Evolutionary Biology Research Programme, University of Helsinki, FI-00014 Helsinki, Finland. [5] Via Barge 3, 10139 Torino, Italy. [6] VVE Workgroup Butterflies, Diamantstraat 4, 8900 Ieper, Belgium. [7] Barrett's Close, Stonesfield, Oxon OX29 8PW, UK. [8] Naturwissenschaftliche Sammlungen, Sammlungs- und Forschungszentrum, Tiroler Landesmuseen, 6060 Hall in Tirol, Austria. [9] Centre for Biodiversity Genomics, University of Guelph, Guelph, ON N1G 2W1, Canada. [10] These authors contributed equally: Vlad Dincă, Leonardo Dapporto. ✉email: vlad.e.dinca@gmail.com

DNA barcoding, the use of short, standardised genomic regions to facilitate species identification and discovery[1], has revolutionised the study of biodiversity. This approach is particularly effective for animals, where the 5' region of mitochondrial cytochrome *c* oxidase subunit 1 (*COI*) aids both specimen identification and species discovery, while also revealing phylogeographic and other eco-evolutionary processes (e.g.[2–6]). Motivated by these applications and aided by cost-effective protocols, 9.1 million DNA barcodes were available on the Barcode of Life Data System[7] by January 2021. As new DNA sequencing platforms reduce analytical costs and allow millions of specimens to be processed annually by a single instrument[8], it is certain that DNA barcode coverage is poised for rapid expansion.

Despite its diverse applications[9], most DNA barcode libraries provide coverage for small geographic areas and/or for narrow taxonomic assemblages. Only few studies have evaluated species-rich taxonomic groups across large areas (e.g.[2,6,10,11]). Yet a key step towards the ultimate goal of revealing global biodiversity involves the development of taxonomically curated sequence libraries for major taxonomic assemblages on a continental or global scale. In this context, Lepidoptera, one of the most diverse insect orders, is the taxonomic group which has gained the most intensive analysis with over 1.4 million DNA barcodes (June 2020) and some of the largest regional barcode libraries available (e.g.[4,6,10,12,13]). This fact reflects the wide interest in Lepidoptera, and their long use as a model system for evolutionary biology, as well as a flagship taxon for insect conservation and for assessing the impact of climate change. This is particularly the case for European butterflies as knowledge of their distributions, ecology, and conservation status is arguably unequalled among invertebrates (e.g.[14–16]).

A final continental-scale taxonomic system for a diverse group such as butterflies cannot be based on the DNA barcode region or any single genetic marker. The need for more comprehensive analysis reflects the occurrence of introgression (e.g. *Spialia sertorius-rosae*[17], *Iphiclides podalirius-feisthamelii*[18], *Melitaea phoebe-ornata*[19]), paraphyly (e.g. *Lasiommata* spp[20].) and reproductive compatibility among deeply divergent lineages (e.g. *Pararge aegeria*[21]). The most recent checklist of European butterflies has been assessed based on the consensus among a group of experts[22] and recognises species only when they are characterised by differentiation in multiple markers. For these reasons, the possibility to assign specimens to their species based on DNA barcodes is not always a binary response, but can be regarded as a continuous probability. However, the degree to which identifications based on DNA barcodes are (or are not) a binary response has never been assessed on the continental scale for a large taxonomic group. To make this assessment possible, PROTAX[23] is a method that gives probabilistic taxonomic assignment (i.e. identification) for a sequence based on a given taxonomy and a set of reference sequences that were identified a priori.

Aside from their taxonomic applications, *COI* sequences are widely used in phylogeographic analysis[24]. Many studies have demonstrated that mtDNA shows strong differentiation among populations particularly in weakly dispersive species[6,24,25]. It has also been shown that the high mutation rate of mtDNA[26,27] exposes processes of differentiation and population dynamics in response to Quaternary climate cycles. Quaternary climatic oscillations are considered among the main forces producing the observed genetic differentiation and the current distribution of lineages[14,28] but, only recently, the increased availability of large DNA barcode datasets has allowed comparative phylogeographic assessments (e.g.[6,29] for European butterflies). However, comparative phylogeography is a largely unexplored field of research and basic questions remain unanswered such as the relationship between genetic richness (i.e. number of haplotypes) and the distribution of species. Indeed, the expected relationship between genetic polymorphism and effective population size or some of its proxies has returned contrasting results in previous studies[6,26,27,30–32]. Other fundamental expectations deriving from the glacial-interglacial expansion and contraction paradigms have been demonstrated for the most part in studies involving a single or a few-species. Among them is the so-called "southern genetic richness and northern purity" generated by rapid northward expansions during interglacial periods which expose lineages to founder effects and gene surfing, resulting in lower genetic diversity in Northern Europe[33].

This study develops a high-resolution DNA barcode library for the entire butterfly fauna of Europe, creating, to our knowledge, one of the most taxonomically and geographically comprehensive libraries available for any group of animals. This comprehensive library can be actively integrated into multifaceted research programmes by highlighting patterns of diversity across space and time, helping to focus conservation action on key populations and taxa. A recent study provided considerable DNA barcode coverage for the butterfly fauna of Western Europe[6], but complete DNA barcode libraries have, until now, only been available for certain countries[4,34–37]. Besides providing a resource for future research, the present dataset makes it possible to (1) estimate mitochondrial genetic diversity and determine the pattern of haplotype frequency among species and the relationship between haplotype richness and range size, as a proxy for population size, (2) reveal latitudinal and longitudinal trends in haplotype diversity across Europe, (3) quantify the ability of the current DNA barcode library to identify European butterflies based on the probabilistic PROTAX method, and (4) highlight taxa deserving more intensive study because of their shallow or deep intraspecific genetic divergence.

## Results

**Assessment of dataset quality.** Our dataset included 22,306 DNA barcodes (>600 base pairs) obtained from butterfly specimens identified at the species level based on a combination of external and internal morphology, life history traits, distribution and nuclear and mitochondrial sequences, in this order (see Methods). As a result, 459 species were represented, providing coverage for 97% of the European butterfly fauna (Fig. 1, Supplementary Data 1, 2, and see the Methods section for further details).

The assessment of dataset completeness in terms of mitochondrial genetic diversity was based on 404 species for which at least six *COI* sequences/species were available. Analysis of rarefaction sampling curves revealed that, on average, 62% of the total estimated number of haplotypes was recovered, and that 261 of the 404 species (65%) had more than 50% of their estimated genetic diversity sampled (Fig. 2a, Supplementary Data 3). Recovery values ranged from a low of about 20% of haplotype diversity in 15 species (*Hyponephele lupina* was lowest at 9.5%), while recovery exceeded 80% for 124 species (30.7%) (Fig. 2a, Supplementary Data 3). The total number of estimated haplotypes varied greatly among species, ranging from one in 15 species to over 300 in two species (*Pyronia cecilia* with 331 haplotypes; *Maniola jurtina* with 354 haplotypes) (Supplementary Data 3).

As expected, the number of sequenced specimens was highly correlated with species range reflecting our decision to analyse more specimens of widespread species (Phylogenetic Generalised Least Square model: $n = 385$, $R^2 = 0.714$, estimate = 0.504, Standard Error = 0.0163, $t = 30.889$, $P < 0.001$). When the number of haplotypes (both detected and estimated) was modelled against the range size and number of sequenced

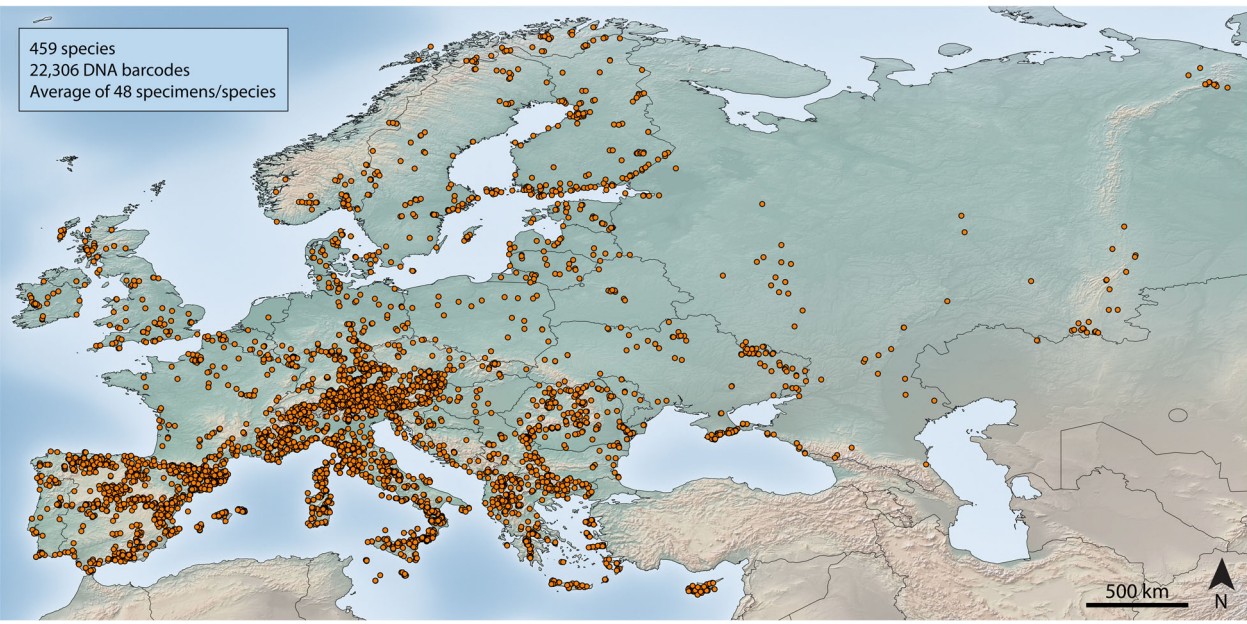

**Fig. 1 Geographic origins for the 22,306 DNA barcodes included in this study.** Denser sampling in Southern Europe reflects the higher number of species in this region. The figure was generated using Quantum GIS 1.8.0 based on a map from Natural Earth (www.naturalearthdata.com).

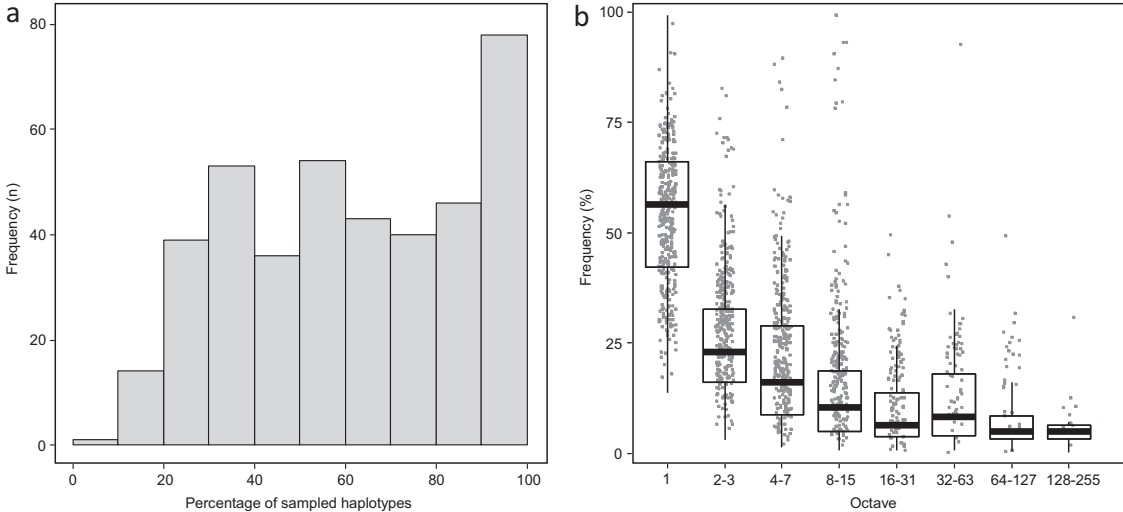

**Fig. 2 Histogram of sampled genetic diversity in European butterflies. a** Observed number of *COI* haplotypes/asymptote for 404 species of European butterflies represented by at least six *COI* sequences; 62% of the estimated haplotype diversity is encompassed by the dataset. **b** Frequency of *COI* haplotypes belonging to the eight log$_2$ classes of abundance for the same 404 species. Most haplotypes were found in single specimens. Boxplots represent median values among species and 25% interquartile ranges.

**Table 1 Results of phylogenetic generalised least squares models relating observed and estimated number of haplotypes with species range and number of sequenced specimens.**

|          | Predictor | Estimate | SE    | t      | P      |
|----------|-----------|----------|-------|--------|--------|
| Observed | Range     | 0.062    | 0.025 | 2.484  | 0.013  |
|          | Specimens | 0.669    | 0.042 | 15.891 | <0.001 |
| Estimated| Range     | 0.125    | 0.038 | 3.280  | 0.001  |
|          | Specimens | 0.778    | 0.064 | 12.173 | <0.001 |

Estimate, standard errors (SE), *t* and *P* values are reported (*n* = 385 species).

specimens, both variables explained a large fraction of variance (observed haplotypes: *n* = 385, adjusted *R*² = 0.747; estimated haplotypes, *n* = 385, *R*² = 0.673; Table 1, Supplementary Fig. 1). A comparison of AICc between the best (multivariate) model and the univariate models revealed the complete model had AICc >2.0 with respect to the highest univariate model (specimens alone in both cases, ΔAICc = 4.14 for observed richness; ΔAICc = 8.66 for estimated richness), thus revealing a strong impact of both predictors in explaining haplotype richness.

In most species, a few common haplotypes representing most of the specimens were joined by many rare haplotypes represented by one or a few individuals. The pattern was evident when median haplotype frequencies were plotted in log$_2$ octaves

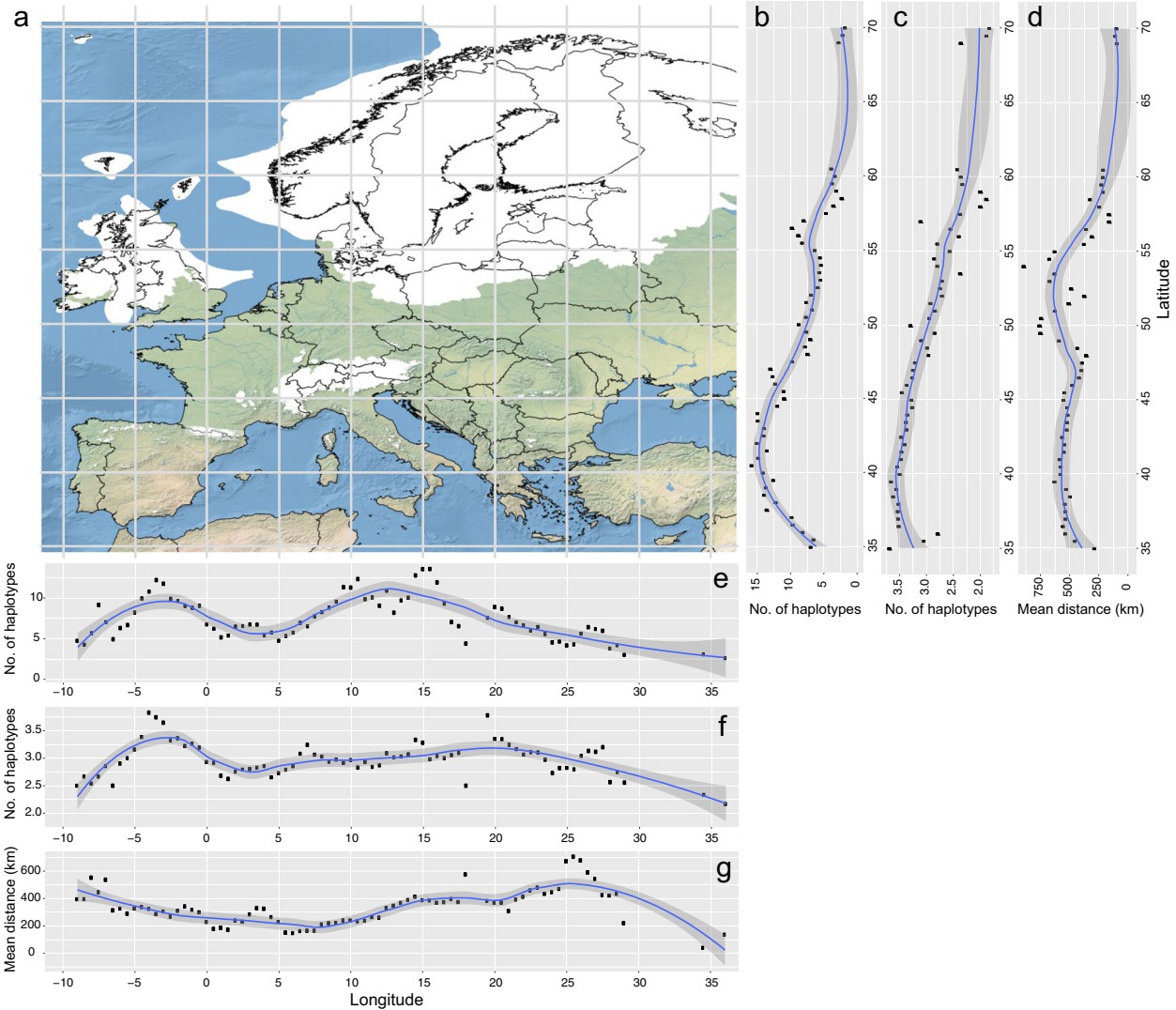

**Fig. 3 Levels of mitochondrial DNA diversity for butterflies across Europe. a** Map of Europe showing the extent of Pleistocene ice sheets (white areas); **b–g** Smoothed Conditional Means plot of haplotype richness using locally estimated scatterplot smoothing with a 95% confidence interval (shaded area); **b** Latitudinal trend in haplotype diversity based on the full dataset; **c** Latitudinal trend in haplotype diversity based on a random homogeneous subsampling of the full dataset; **d** Latitudinal trend of geographic distances among sampled specimens for the full dataset; **e** Longitudinal trend in haplotype diversity based on the full dataset; **f** Longitudinal trend in haplotype diversity based on a random homogeneous subsampling of the full dataset; **g** Longitudinal trend of geographic distances among sampled specimens for the full dataset. Latitude was divided into $n = 65$ belts and longitude was divided into $n = 77$ belts.

of abundance, as haplotypes represented by a single specimen were the most frequent category (Fig. 2b, Supplementary Data 4).

**Mitochondrial genetic diversity across Europe**. The distribution of mtDNA diversity across latitude was not homogeneous (Fig. 3a–d). This fact was demonstrated by comparing the mean number of haplotypes predicted for each species based on rarefaction curve asymptotes with latitude and longitude (included as smoothed predictors in the Generalised Additive Mixed Models controlling for spatial autocorrelation) and with distance between sampled specimens as a parametric predictor (Supplementary Data 5, 6). Latitude and longitude showed significant effects (Supplementary Data 7) while distances between the sampled specimens, did not (Supplementary Data 8).

A Smoothed Conditional Means plot using locally estimated scatterplot smoothing (loess, with α = 0.4) made it possible to inspect the trend of these significant relationships. This analysis showed that estimated haplotype richness was highest (>12

haplotypes per species) at latitudes ranging approximately between 38° and 47°. This belt included most of the large southern European peninsulas, as well as the Pyrenees, southern Alps, the Balkans, and the southern Carpathians. More southerly regions of Europe (<38°) displayed fewer haplotypes and a similar decline was evident at >47° with most regions in Scandinavia displaying less than five haplotypes per species. A secondary peak in haplotype richness around 55° may reflect stochasticity or the presence of high geomorphological complexity reflecting marine barriers between major land masses (Ireland, Britain, Central Europe, Scandinavian Peninsula).

When this analysis was rerun based on a random subset of specimens to mitigate the potential effects of unequal sampling intensity (see Methods), the Generalised Additive Mixed Models still revealed a significant effect for latitude and longitude and no effect for distance among specimens (Supplementary Data 7, 8). The loess regression plot showed that the decrease in haplotype diversity at high latitudes was maintained while the secondary peak around 55° was diminished and the southernmost parts of

Europe were now among the regions with the highest mitochondrial diversity (Fig. 3c).

The full dataset was also used to assess shifts in mitochondrial diversity with longitude. This analysis showed the highest diversity in two belts, approximately −5° to 0° and 9° to 16°, (Fig. 3e). These belts largely correspond to the Iberian and Italian Peninsulas, two of the main glacial refugia in Europe. When the analysis considered a random subset of specimens, the Iberian belt of high diversity was retained, followed by a relatively homogeneous distribution of mitochondrial diversity to the east, characterised by a less pronounced increase peaking around 20° longitude, roughly corresponding to the western Balkan Peninsula (Fig. 3f).

Similar to latitude, this pattern did not appear to be explained by the geographic distance between the sampled specimens, since the average distances between conspecific specimens were highest in the extreme west (−9° to −7°) and in Eastern Europe (24° to 27°) (Fig. 3g).

**Identification of European butterflies using DNA barcodes.** PROTAX analyses showed that, when averaged across all DNA barcodes in our dataset, the average probability of assigning a specimen to the correct species (using current taxonomy[22] as the point of reference) was 95.3% (Supplementary Data 9). This average value reflected the fact that 17,917 specimens were identified with a probability of 100% while the probability of correct identification for 871 specimens was less than 50%, including 13 records with zero probability (Supplementary Data 9).

When the average probability of correct identification was examined for each of the 451 species, 18 (4%) had a probability of correct identification <50% (minimum 14.8% for *Hipparchia volgensis*) while 370 (82%) species had a probability >95% and 327 (72.5%) had a probability of 100% (Fig. 4, Supplementary Data 10). These analyses excluded the eight taxa represented by

singletons as PROTAX probabilities could not be inferred for them (Supplementary Data 9).

The relationship between the distance to the nearest neighbour (i.e. minimum genetic distance between specimens of a particular species to any other species in the dataset) and the average probability of correct taxonomic assignment for each species revealed that PROTAX probabilities were at least 99.9% when the distance to nearest neighbour was >0.61% (4 substitutions) (Fig. 4).

A BIN (Barcode Index Number) analysis revealed that the 459 species of European butterflies from our dataset were assigned to 441 BINs (360 concordant BINs, 16 singleton BINs and 65 discordant BINs) (Supplementary Data 11–13). The 65 discordant BINs involved 168 species, with up to 11 species sharing a BIN (Supplementary Data 13).

**Cases of barcode sharing, low distance to nearest neighbour, or high intraspecific variability.** Among the 22,074 specimens identified to a species level based on our integrative approach (459 species), 29 cases of barcode sharing were detected involving 69 species (15% of the species) (Supplementary Data 14, 15). These 29 cases included 22 pairs, five triplets, one tetrad, and one case involving six species. None of the 69 species involved in barcode sharing had an average PROTAX probability of correct identification of 100%, but values varied from 14.8% (*H. volgensis*) to 99.6% (*Melitaea celadussa*). Based on additional evidence for these cases, we estimate that 65.5% (19 cases) of the 29 cases of barcode sharing reflect operational factors (i.e. further taxonomic research is needed) while 34.5% (10 cases) are most likely due to biological processes (e.g. mitochondrial introgression) (Supplementary Data 15).

Another category involved species that did not share DNA barcodes, but whose PROTAX probabilities of correct taxonomic assignment was <99.9% (40.1%–99.8%). This category included 50 species (10.9% of the total), all displaying <0.61% distance to

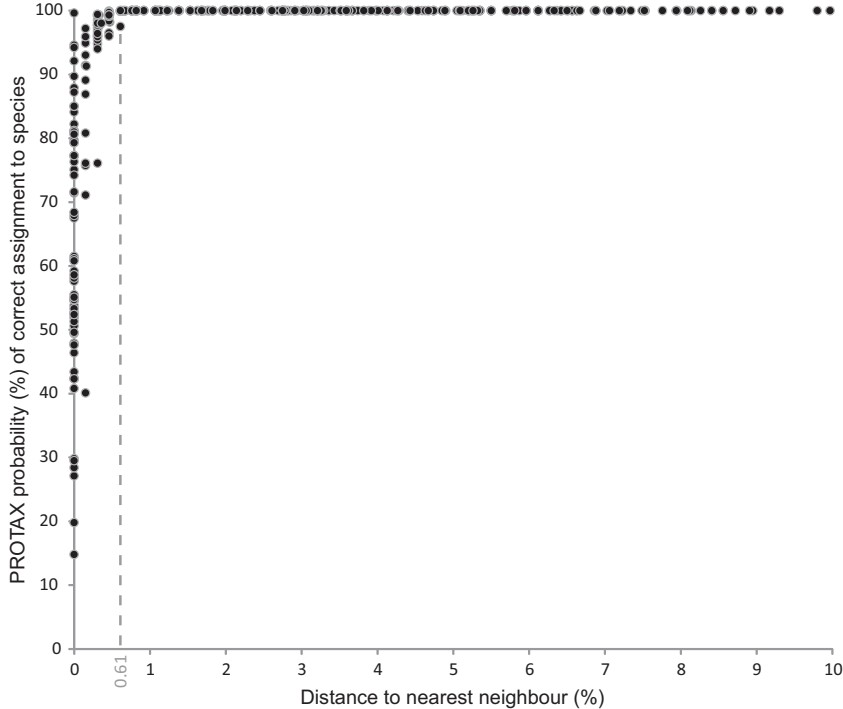

**Fig. 4 Relationship between distances to nearest neighbour and PROTAX probabilities of correct assignment of specimens to species.** Distances to nearest neighbour represent the minimum genetic distance between specimens of a given species from any other species in the dataset (*x*-axis). PROTAX probabilities of correct assignment to species (*y*-axis), represent the average probability for each species (*n* = 451 species).

their nearest neighbour (Fig. 4, Supplementary Data 14, 15). More precisely, only one species pair (*Erebia meolans* + *E. palarica*) displayed a distance to the nearest neighbour of 0.61%, while the rest diverged by <0.5%.

Conversely, 12.2% of the species (56 of 459) had a maximum intraspecific p-distance (proportion of nucleotide sites that differ between two sequences) of >2.5%, about six times higher than the average mean intraspecific distance (0.41%) for all 459 species and over twice as high as the average maximum intraspecific distance (1.19%). Among these species, the highest intraspecific p-distance was detected in *Colias palaeno* (7.80%) and *Melitaea didyma* (7.87%) (Supplementary Data 14, 16).

## Discussion

Comprehensive sampling, both in taxon and geographic coverage, has been highlighted as critical for the effective application of DNA barcoding to species identification and to assessments of intraspecific genetic diversity (e.g.[38]). To our knowledge, our sampling is among the most comprehensive available for any taxonomic group at a continental scale (Fig. 1). Although we recovered >80% of the expected haplotype diversity in more than 30% of the species (Fig. 2a), an average sample size of 48 specimens per species only retrieved a mean of 62% of the total estimated haplotype diversity (Fig. 2a, Supplementary Data 2). This gap in coverage was largely explained by the fact that haplotype frequencies were not normally distributed – instead there were a few common haplotypes and many rare ones (Fig. 2b), making it nearly impossible to capture all diversity. This highly skewed abundance of *COI* haplotypes has methodological implications for research aiming to estimate mitochondrial DNA diversity[39]. The factors underlying this phenomenon require additional study. It is possible that recurrent selective sweeps result in a low number of prevalent haplotypes, followed by a myriad of low-frequency variants created by subsequent mutations. Other potential causes (not necessarily mutually exclusive) might include processes such as low selection on most individual mutations that tend to accumulate, or frequent rate of mutation followed by rare fixation.

Another notable finding is the high variability in the extent of mtDNA diversity for European butterflies (Supplementary Data 3). This variability is explained by species range size even after controlling for the larger number of specimens sequenced for taxa with broad distributions. This result confirms expectations based on at least two strong theoretical predictions: (1) neutral genetic diversity should equate to the product of mutation rate and effective population size and (2) more widely distributed species are more likely to undergo allopatric genetic differentiation in isolated areas. Nevertheless, prior studies did not detect a correlation between mtDNA polymorphism and proxy measures of population size[26,30,31] and a recent study correlating haplotype diversity with different traits of European butterflies, including range size, did not show a positive relationship[6]. The difference can be due to the use of number of haplotypes in this study and of a different measure for haplotype diversity in reference[6], and/or by the lower number of predictors (and higher number of species) considered in this study compared to reference[6], which could have rendered a higher power to the analysis conducted here.

Despite the highly significant relationship between range size and haplotype richness, some common, widespread species (putatively having large population size) possess little variation. However, the converse was never true, no species with a narrow range possessed a high number of haplotypes (see Supplementary Fig. 1 and Supplementary Data 14). Widespread species with low mtDNA diversity include *Gonepteryx cleopatra* (2 estimated haplotypes), *Colias hyale* and *Hamearis lucina* (each with 3 haplotypes), *Araschnia levana* (4 estimated haplotypes), and *Nymphalis antiopa* (4 estimated haplotypes). Their low diversity may reflect a recent post-glacial expansion of small populations or the occurrence of recent selective sweeps that erased most diversity. One main factor inducing mitochondrial selective sweeps may be the bacterium *Wolbachia*, which can manipulate the reproductive dynamics of its hosts, often through cytoplasmic incompatibility when sperm from infected males cannot fertilise eggs unless females are infected by the same *Wolbachia* strain (e.g.[40]). In such situations, *Wolbachia* can cause a selective sweep of the mtDNA variant carried by the infected individuals (e.g.[41]). Several cases have been revealed where *Wolbachia* infections are correlated with mtDNA genetic structure in European butterflies (e.g.[17,18,42,43]).

When examined across the continent, levels of genetic diversity in European butterflies were heterogeneous with a clear decrease above about 47°N (Fig. 3a–d). According to numerous case studies, a massive ice sheet covered much of Europe above 50°N[44] forcing most species to retreat to three glacial refugia located in southern peninsulas (Iberia, Italy, Balkans - also highlighted in the longitudinal trends, Fig. 3) where they diverged[14,28]. Our observation that European butterflies show a gradual south-north decrease in genetic diversity represents the first general evidence of southern genetic richness and northern purity for a large taxonomic group at a continental scale (Fig. 3). Furthermore, it is possible that ongoing diversification and more limited gene flow in geographically more heterogeneous southern Europe (high mountains separated by lowland areas or sea straits) also contributed to an increased genetic diversity compared to the more uniform northern Europe. Interestingly, although some of the main southern mountain chains were ice-covered during glacial maxima, they are among the areas with highest genetic diversity, likely because they provided refugia for cold-adapted species during the current interglacial period and because of their proximity to the southern European refugia[45].

Most studies evaluating the effectiveness of DNA barcoding for identifying specimens to a species level have used distance-based, monophyly-based or, less frequently, character-based methods. The distance-based method uses genetic divergence values as a criterion and relies upon the "barcoding gap", i.e. the gap between intra- and interspecific variability[38]. Alternatively, success can be judged by monophyly on a phylogenetic tree, or by employing the character-based approach, which relies on diagnostic combinations of nucleotides for species discrimination (e.g.[46,47]). Regardless of their theoretical and practical limitations which have been the subject of various reviews (e.g.[48]), these methods assess identification success in absolute terms ("yes" or "no"). For example, cases of barcode sharing between species represent instances of barcoding failure, although the probability of such failures (which depends on the frequency of barcode sharing) is not considered.

Based on a set of reference sequences, in this study identified a priori to their species based on a combined collection of available data (see Methods), PROTAX[23] reports taxon membership for unknown specimens as probabilities. The probability informs about the (un)certainty of the taxonomic assignment. This method does not require any a priori defined similarity threshold or barcode gap in order to report taxon memberships of query sequences. Neither does it assume that the taxonomic units should be separable. When comparing the query sequence against a taxon, PROTAX calculates genetic distances between the query sequence and all reference sequences of the taxon. It utilises both minimum and average distance as predictors when assigning the taxon probability. By means of the average distance it considers the haplotype frequency of the reference sequences. Thus, PRO-TAX can deal with any (small or large) interspecific genetic

distance above zero, whereas some procedures defining de novo groups based exclusively on single-marker genetic distances or phylogenetic tree (like the BIN approach) can produce cases of over-splitting or lumping because coalescence times can largely vary among taxa (e.g.[20] for tribes of Nymphalidae). Because PROTAX also takes into account haplotype frequency, lack of monophyly due to low-frequency haplotypes (e.g. rare cases of introgression) has a low impact on the performance of this method. PROTAX accuracy increases when taxonomic knowledge and coverage, as well as sampling quality, are high[49] and it relies on assumptions that should be revised when taxonomic knowledge progresses.

Our comprehensive DNA barcode library of European butterflies coupled with PROTAX should represent a powerful method of attributing specimens to their correct species. Although PROTAX generates continuous probabilities for the attribution of specimens to a given species, its application to our extensive dataset appeared almost dichotomous (Fig. 4), confirming the power of DNA barcodes to distinguish even genetically similar taxa. In fact, the PROTAX probabilities of successful species assignment abruptly increased to >99.9% when levels of interspecific divergence were above 0.61%. This latter value is certainly at the lower end of interspecific divergence values based on past estimates of interspecific divergence values for COI in Lepidoptera (e.g.[4,10,34,37,50]). It is also worth emphasising that certain species which share barcodes (i.e. distance to nearest neighbour = 0) often have high probabilities (>85%) of correct taxonomic assignment (Fig. 4, Supplementary Data 10) because barcode sharing often involves only few, presumably introgressed, specimens. This result exemplifies how PROTAX can reveal subtler patterns than when absolute criteria are employed to quantify identification success, but also highlights the necessity for a comprehensive dataset.

When using the BIN system[51], and comparing the resulting groups with the identification of specimens made in this study[22], 168 species (36.6% of the species in the dataset) were involved in BIN discordance (i.e. at least two species sharing the same BIN), suggesting that barcoding identification effectiveness using BINs is 63.4%. This reflects the fact that several species of European butterflies display low COI minimum interspecific distances or share barcodes (even if the latter sometimes occurs at low frequency), leading to BIN sharing (Supplementary Data 13, 15).

When viewed from the perspective of absolute certainty, identification success to a species level cannot be higher than 85% for European butterflies because 15% of the species show barcode sharing (Supplementary Data 15). However, the assessment based on probabilities provides an evaluation of each case by considering the frequency of barcode sharing, which was often limited to a few specimens. It should also be noted that in all cases of barcode sharing, identification is still possible to a few closely related species (typically only two taxa, Supplementary Data 15). Moreover, since several cases involve allopatric or parapatric taxa[15], filtering by sampling site can often allow reliable species assignments enhancing the capability of DNA barcodes to deliver a reliable identification.

As an example, Melanargia galathea, M. lachesis and M. larissa, although differentiated by morphological and allozyme markers, only differ by a few mutations in COI (distances to nearest neighbour < 1%)[52,53]. Nevertheless, they are reliably identified by PROTAX (identification > 99.3% for the three species, Supplementary Data 10), while they are recognised as a single entity by the BIN analysis (Supplementary Data 13). Other species that display low genetic differentiation (distances to nearest neighbour < 1%) while not sharing barcode haplotypes are placed in the same BIN (e.g. Polyommatus icarus and P. eros, Pyrgus cacaliae and P. andromedae, Gonepteryx cleopatra and G.

farinosa, see Supplementary Data 15), but PROTAX attributed 100% of specimens to their correct species. Since several of these species are widespread and common in Europe, we conclude that PROTAX coupled with the current comprehensive and taxonomically-assigned DNA barcode library can correctly identify a large fraction of diversity that would remain undetermined by using other methods.

As mentioned above, 15% of European butterfly species are involved in barcode sharing (Supplementary Data 15), a value considerably higher than values reported for any area of the continent (e.g.[4,34,36,37]). This increase is most likely due to the higher taxon and geographic coverage of our dataset. The percentage of barcode sharing in European butterflies is also high when compared with the few studies that provide comprehensive, continental-scale datasets for other taxa. For example, a study on North American birds[2] examined 93% of the avifauna of USA and Canada with an average of 4.1 specimens/species and with 15% of the species represented by singletons. In this case, only 0.3% (2 species) out of 643 species analysed shared DNA barcodes. For the North American Noctuoidea[10], taxon coverage was 97.3%, with an average of 19.5 specimens/species. In this case, 10.3% (158 species) of the 1541 species analysed shared DNA barcodes. These results agree with our observation that when barcode sharing occurs, it is often so uncommon that it would be overlooked with smaller sample sizes.

Barcode sharing can be caused by several factors, such as introgression following hybridisation, incomplete lineage sorting, or by operational factors (taxonomic oversplitting, identification uncertainties). Assessing the exact contribution of each of these factors to European butterflies will require exhaustive studies. We estimate that about two thirds of the cases of barcode sharing reflect operational factors (i.e. taxonomic research is needed to properly define species boundaries), while the rest are likely due to biological processes such as introgression (Supplementary Data 15). The presumed high impact of operational factors agrees with an earlier study that examined almost 5000 species of European Lepidoptera[54], where 58.6% of the non-monophyletic species were likely caused by operational factors.

Among operational factors affecting identification, it should be noted that some European taxa listed by Wiemers and collaborators[22] are not only morphologically similar, but they often occur in allopatry or parapatry, sometimes with known hybrid zones (e.g. Melitaea athalia and M. celadussa[55]). Such conditions make species delimitation both challenging and dependent on the species concept adopted[29,56]. Some genera also have notoriously difficult taxonomy (e.g. Pyrgus, Hipparchia, Melitaea, Lysandra)[15,19,57,58], and much work is needed to better understand their speciation patterns. On the other hand, introgression undoubtedly explains some cases of DNA barcode sharing in European butterflies (e.g.[18,19,58]). This conclusion agrees with estimates that about 16% of European butterfly species hybridise in the wild, with about half of these cases producing fertile hybrids[59].

Apart from the cases of DNA barcode sharing, PROTAX probabilities of correct taxonomic assignment under 99.9% were only detected for species with very low distances to their nearest neighbour (usually under 0.5% and no more than 0.61%) (Fig. 4., Supplementary Data 15). These species represent 10.9% of the dataset and, although further research is needed, some cases are likely to reflect operational factors (especially unresolved taxonomy), similar to the cases of barcode sharing. Indeed, the taxonomic status of some of these species is controversial with certain authors treating them as subspecies (e.g. Satyrus virbius)[22]. It is also likely that some cases reflect incipient speciation, a challenging situation for taxonomy. However, some species pairs involve well-differentiated taxa (e.g. Colias palaeno - C. tyche, Brenthis daphne - B. ino) (e.g.[60].), so the low mtDNA divergence

may reflect processes such as incomplete lineage sorting or recent introgression.

DNA barcoding studies also frequently report cases of species showing deep *COI* divergence that may reflect cryptic diversity (two or more morphologically similar species currently classified as one)[61]. This layer of biodiversity appears to be more prevalent than previously thought (e.g.[62]) with several recent cases documented in European butterflies, many first highlighted by differentiation in mtDNA (e.g.[17,34,63,64]).

There is no hard threshold that represents high intraspecific divergence in mtDNA, but minimum intraspecific distances higher than 2–2.5% are often viewed as indicative of potential cryptic diversity in Lepidoptera (e.g.[4,10,37]). To provide an overview for European butterflies, we used a threshold criterion of 2.5% maximum intraspecific divergence (Supplementary Data 16), a value exceeded by 12.2% of the species in our dataset versus 7.2% of the species when the threshold was increased to 3%. In principle, the examination of variation in the nuclear genome should help clarify these cases, but the outcomes can be complex. For example, recent studies employed double digest RAD-sequencing (ddRADseq) to investigate two European butterfly species with high mtDNA intraspecific divergence (*Thymelicus sylvestris* and *M. didyma*). In *T. sylvestris*, mitonuclear discordance was detected, and the ddRADseq data did not suggest the presence of cryptic species[65]. By contrast, *M. didyma* displayed mitonuclear discordance, but the ddRADseq data recovered five well-differentiated allopatric or parapatric lineages, highlighting the difficulties of deciding on taxonomic status given the continuous nature of the speciation process[56]. On the other hand, overlooked species have been documented despite barcode sharing, most likely due to genetic introgression (e.g.[18,19]). Certain sister species display low minimum interspecific divergence, with some of the best-known cases in European butterflies being those of *Leptidea*[66,67], where *L. sinapis* and *L. reali* differ by less than 1% in DNA barcodes, and *Melanargia galathea* and *M. lachesis*, which differ by just three substitutions[52].

In conclusion, the DNA barcode library for European butterflies generated in this study is one of the most comprehensive for any taxonomic group at a continental scale. When used with PROTAX, this library is highly effective (>95%) in assigning unknown *COI* sequences to their correct species, as defined according to current taxonomy. Construction of the library also allowed estimation of the total mitochondrial genetic diversity present in each European butterfly species while also providing a continental-scale perspective on patterning of genetic diversity for a taxonomically diverse group.

This library will additionally stimulate research to improve understanding of mechanisms shaping genetic differentiation, regardless of taxonomic conclusions. Deep intraspecific splits can also be regarded as evolutionary significant units that will help to focus conservation efforts[68]. In this respect, the results of this study will aid nature conservation by providing broad context genetic information that can help prioritising management actions[43].

## Methods

**Sampling and collection data**. The present dataset includes 22,306 *COI* sequences from 459 species of European butterflies (Supplementary Data 1). This represents 97% of the 473 species occurring in Europe (Supplementary Data 2), not including the Macaronesian islands, where 23 additional species occur[22]. The number of sequences per species ranged from 1 (8 species) to over 200 (12 species), with an average of 48 sequences per species. Among this total, 232 specimens were only identified to genus level (see below), meaning that 22,074 specimens were identified to species level (Supplementary Data 1). The limits of the continent follow reference[22] with the eastern limit set at 66.5° longitude, along the Urals (Fig. 1). The Macaronesian islands were excluded because they are not representative of the European fauna from a biogeographic point of view. 5015 *COI* sequences were generated for this study, while the remainder originated from public data available on the Barcode of Life Data System (BOLD)[7], the main sources being[4,6,12,29,34–36,50,54].

We preferentially employed DNA barcodes from BOLD because this platform facilitates both verification of specimen identifications and, if necessary, assessments of the quality of electropherograms.

To provide a good representation of intraspecific *COI* variability, sampling was designed to cover, as much as possible, the known range of each species of European butterfly so hundreds of localities were sampled across the continent (Fig. 1).

**Specimen identification**. Efforts were made to ensure the correct identification of each specimen in the dataset. Following a recent checklist of European butterflies[22] and available information in literature about morphology, life history, distribution and genetic markers, we applied an integrative approach. In practice, when information about external and/or internal (genitalia) morphology was sufficient to identify a specimen, we attributed the specimen to a species. In problematical instances (e.g. cryptic species), information relating to species ecology and life history (e.g. larval food plant, habitat, phenology) was used; if this was still insufficient we compared the collection site with the known distribution of species; then we used information from nuclear DNA (when available). Therefore, nearly all specimens were identified prior to obtaining their DNA barcodes. Mitochondrial DNA was used as a last resort (always in combination with other data, e.g. distribution) in only a small number of specimens, including taxa for which previous studies have shown an apparently perfect correspondence between DNA barcodes and species delimitation based on other criteria (e.g. *Leptidea sinapis*, *L. reali*, *L. juvernica*). Genitalia were examined for 1490 specimens where external features were considered insufficient for reliable identification. The genitalia were processed as follows: maceration in 10% potassium hydroxide, cleaning and examination under a stereomicroscope and storage in tubes with glycerine or on permanent microscope slides. Despite these efforts, genitalia were not available for examination in some cases or they were not examined due to time constraints. In other cases, genitalia morphology, or other characters, did not provide sufficient information for reliable identification. For these reasons, 232 specimens were identified only to the genus level (Supplementary Data 1and further information in the Supplementary Methods).

Following morphological analyses (especially genitalia morphology), a few specimens turned out to be initially misidentified and these cases were corrected in the final dataset (e.g. in genera such as *Plebejus*, *Melitaea* or *Hipparchia*, which include several morphologically very similar species).

Taxonomy and nomenclature followed a recently updated checklist of European butterflies[22].

**Analyses of DNA sequences**. Further information regarding the methodology employed is available in the Supplementary Methods.

Most DNA barcodes used in our dataset were obtained following standard protocols for Lepidoptera[69]. A number of sequences were generated in the Butterfly Diversity and Evolution Lab at the Institute of Evolutionary Biology, Barcelona (Spain). We retained only *COI* sequences of at least 600 base pairs.

To facilitate the visualisation of genetic distances, a neighbour-joining tree of the 22,306 *COI* sequences was built using BOLD, based on uncorrected p-distances[70] (Supplementary Data 14). Other distance-based analyses (e.g. distances to nearest neighbour) were also run using BOLD.

**Statistics and reproducibility**. We assessed the completeness of our sampling of haplotypes for all species with at least six specimens. The abundance of each haplotype was scored for each species and rarefaction curves were then calculated by plotting the number of individuals versus the number of haplotypes using the iNEXT function of the iNEXT R package. iNEXT also calculates asymptotes predicting the expected number of elements in rarefaction curve analysis[71]. The expected (asymptotic) number of haplotypes was compared with the observed number to quantify the level of sampling completeness.

For each species, haplotypes were divided into eight geometric classes of abundance: 1, 2–3, 4–7, 8–15, 16–31, 32–63, 64–127, 128–255. Subsequently, we calculated the relative frequency of haplotypes in each class and then computed the mean frequency of haplotypes for all species in each class, as well as the standard deviation.

Range size has been used as a proxy for both effective population size and as a proxy for the likelihood that a species possesses disjunct populations[6]. Range size estimates for most European butterflies are available in the CLIMBER dataset ([72] and update in[73]) as the number of 30 × 30 km square cells occupied in Europe. A relationship between haplotype richness and range size can be due to the fact that species with a wider range are also more heavily sampled although the asymptotic values obtained by iNEXT should be relatively independent of the number of sequenced specimens. We tested the relationships between haplotype richness (observed and expected), number of specimens sequenced and range size (all log transformed to improve normality) by using Phylogenetic Generalised Least Square models. As a reference phylogenetic tree, we used the recently published time-calibrated tree for European butterflies[74] and Pagel's lambda as a model for the phylogenetic covariance of residuals as implemented in the function "pgls" of the R package "caper".

To understand if range size makes an independent contribution to haplotype richness, we calculated AIC values for the multiple model (range + specimens) and for the two univariate models (range, specimens) using the "MuMIn" R package. This function enables model comparisons by using the corrected Akaike Information Criterion (AICc). Models with a difference in AIC with the best model (ΔAICc) higher than two are not considered to be equally parsimonious[75].

To compare haplotype diversity at different latitudes we proceeded as follows. We identified 65 partitions separated by 0.5° of latitude and ranging from 35° to 70°. For each partition, we gathered all specimens collected within a range of 1° latitude above and below it. The specimens were divided into species and counted. When a given species was represented by at least six specimens for a given parallel, we included it in the analysis. Since the number of individuals across parallels differed, we scored each selected species for the number of haplotypes present at a given parallel in two ways: (i) by means of rarefaction-extrapolation curves as implemented in iNEXT using the maximum number of sampled specimens of this species across parallels as the extrapolation limit and (ii) for each species in each parallel, we used a subset as large as the minimum number (>5) of specimens analysed in the least sampled parallel. The first method allowed evaluation of the theoretical richness of haplotypes across latitudes, while the second allowed for a standardised evaluation independent of variation in sample size. For each parallel, the mean value across species was computed. Higher haplotype richness may be generated by greater geographic distance among specimens for a species. For this reason, we also computed the mean geographic distances among specimens of each species for each parallel and plotted it for comparison. The relative effect of latitude and longitude and of average distances between specimens for the full and the subsampled data were tested in a series of four Generalised Additive Mixed Models (GAMM). We used the "gamm" function of the "mgcv" R package by including haplotype diversity (full dataset latitude, subsample latitude, full dataset longitude, subsample longitude) as a response variable, latitude or longitude as a smoothed predictor ($k = 10$) and average distances as a parametric predictor. Use of GAMM made it possible to control for spatial autocorrelation of data.

The same methodology used for latitude was also applied for longitude, but in the latter case, we identified 77 partitions separated by 0.5° of longitude and ranging from −9° to 36° (there were no usable meridians east of 36°).

In case of a significant trend between haplotype richness and latitude-longitude, the relationship was inspected and plotted using smoothed conditional means plots with the method of locally estimated scatterplot smoothing (loess, with α = 0.4). Loess graphs were plotted using the "ggplot2" R package.

The extent of the Pleistocene ice sheets illustrated in Fig. 3a was inferred based on reference[76].

We used PROTAX[23] to study the confidence of taxonomic assignments for all DNA barcode sequences. PROTAX is a hierarchical classifier that gives a probabilistic taxon assignment to a query sequence. It relies on a predefined taxonomy and on a set of reference sequences for the taxa in it. The classification starts at the root node of the taxonomy tree where a query sequence belongs with the probability of one (100%). In our model, the root node represents the order Lepidoptera. In order to reach the species level, the classification process considers four levels of the taxonomy: (1) from order to family, (2) from family to subfamily, (3) from subfamily to genus, and (4) from genus to species. Each taxonomic node partitions the probability to its descendants by means of a multinomial regression model. As predictors in each model, we used the smallest p-distance and the average p-distance between a query sequence and the set of reference sequences for each taxon. Model parameters were obtained using Markov Chain Monte Carlo (MCMC) with adaptive proposal distribution[23]. Prior distributions were zero-mean Gaussians with large standard deviations (sd = 100) compared to the values of predictors that had been scaled to have zero mean and unit variance. Adaptive proposal was constructed so that the targeted acceptance ratio was 0.44. 1000 MCMC iterations were run after which the proposal distribution was fixed and another 1000 iterations were performed. Maximum a posteriori (MAP) estimates were obtained from the second half of the iterations.

We used all 22,074 DNA barcodes in our library (identified to species level) as a reference set. After training the model, the taxonomic assignment to a species level for each of the 22,074 barcode sequences was obtained in a leave-one-out fashion, i.e. a barcode was not allowed to be used as a reference sequence for its own classification. PROTAX gives a probabilistic assignment to all nodes in the taxonomy for each query sequence, but in the present analysis, we only used the probability corresponding to species. Of the 22,074 specimens included in the analysis, eight species were represented by singletons so their probability of assignment to the correct species was zero. Therefore, the interpretation of results was based on 22,066 specimens and 451 species (Supplementary Data 9, 10).

Additionally, we also used BOLD to assign each sequence to a BIN[51].

**Reporting summary**. Further information on research design is available in the Nature Research Reporting Summary linked to this article.

## Data availability

All sequences in this study have been submitted to GenBank (accession codes MW498979 - MW503694) and, together with associated information, are publicly available in the dataset DS-EUGENMAP[77] (dx.doi.org/10.5883/DS-EUGENMAP) on BOLD at www.boldsystems.org.

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

## Acknowledgements

>We are grateful to the many colleagues who supported the EUGENMAP project by providing specimens or relevant information. We thank authorities in all European countries that supported the project and provided permits for sample acquisition. Axel Hausmann kindly provided access to all butterfly data from the FBLRH project on BOLD. Support for this research was provided by a Marie Curie International Outgoing Fellowship within the 7th European Community Framework Programme (project no. 625997) and by the Academy of Finland to V.D. (Academy Research Fellow, decision no. 328895), as well as by projects CGL2010–21226/BOS and CGL2013–48277-P (Spanish Ministerio de Economía y Competitividad), CGL2016–76322 (AEI/FEDER, UE), PID2019–107078GB-I00 (AEI/10.13039/501100011033) and 2017-SGR-991 (Generalitat de Catalunya) to R.Vi. and by the projects "Ricerca e conservazione sui lepidotteri diurni di sei Parchi Nazionali dell'Appennino centro-settentrionale", "Ricerca e conservazione sugli Impollinatori dell'Arcipelago Toscano e divulgazione sui Lepidotteri del parco" and "Servizio di attuazione delle azioni per la protezione degli impollinatori e diffusione dell'entomofauna del Parco Nazionale dell'Alta Murgia" to L.D. P.Hu. is indebted to the Promotion of Educational Policies, University and Research Department of the Autonomous Province of Bolzano – South Tyrol for funding. FinBOL was funded by the

Academy of Finland (through FinBIF), Kone Foundation and Finnish Cultural Foundation. Support to P.He. from the Canada Foundation for Innovation, Ontario Genomics, Genome Canada, and the Ontario Ministry of Research and Innovation aided sequence analysis.

## Author contributions

V.D., R.Vi. and L.D. designed the study, V.D. and L.D. performed genetic analyses, P.S. performed PROTAX analyses, V.D., S.C., L.D., R.Vo., R.Vi., M.G.-P., M.M., P.Hu. and P. He. supported the sampling and DNA sequencing effort. V.D., L.D. and R.Vi. wrote the manuscript with contribution from P.He., M.M., P.S., R.Vo., S.C., M.G.-P. and P.Hu.

## Competing interests

The authors declare no competing interests.
