## [Peer Review File · Communications Biology]

Reviewers' comments:

Reviewer #1 (Remarks to the Author):

This is an impressive study that takes DNA barcoding to the next level of resolution at the individual genotype, rather than species identification. About a quarter of the 20000+ sequences were newly generated, resulting in a dataset that is largely complete for all species of butterfly across Europe. This allows analyses of patterns of diversity at the levels of mitochondrial haplotypes, which the study exploits nicely to assess latitudinal and longitudinal gradients. This work reveals the expected patterns of high diversity near the southern part of the continent, and some variation longitudinally that were explained by differences in the land area correlated to the Mediterranean peninsulas. We also obtain information on the number of different haplotypes in each species, revealing great differences ranging from a few to several hundred in some cases, and the distribution of common and rare haplotypes. I think it is an excellent resource and a beautiful example of the power of dense mitochondrial barcoding.

However, the study has some critical flaws that should be addressed. A key conclusion is that mitochondrial variation is not reflecting effective population size. This is supported with a few casual examples, e.g. the fact that fairly widespread species like *Gonepteryx cleopatra* exhibit only a few haplotypes. There is no analysis of any kind beyond these anecdotal observations. Perhaps one could make the opposite argument, i.e. that population size is correlated with effective population size. For example, the widespread *Pyronia cecilia* exhibits 331 haplotypes. But this, again, is just another anecdote. This type of conclusion needs more critical thinking and an analytical framework based in coalescence theory. As it stands, it simply gives ammunition to the idea that DNA barcoding and mitochondrial markers in general are inferior to population studies using nuclear genes and ultimately will restrict its wider use.

Second, the analysis of the taxonomy using PROTAX is a curious approach. The authors aim to test the placement of individual barcode sequences into the named taxa, finding that this works most of the time when these groups are discrete and genetically distant, but not if they are poorly separated. Based on the description of the algorithm in the Methods section, this is not a surprise at all because the method simply gives a probability of the fit in a certain group based on genetic distances. If the Protax placement isn't correct, this is simply the result of the composition of the groups. The problem can only be addressed with a study of what should be the groups, rather than an algorithm that tests the placement of a sequence into the groups. As the authors are well aware, there are many approaches to address this issue that are widely established in barcoding, including methods based on diagnostic character, tree-based methods, and clustering algorithms such as BIN and ABGD (which are probably closest to what Protax does). Any of these should have been used, but not a method such as Protax that does not address what should be the extent of the groups. As it stands it provides a convoluted analysis of the 'barcoding gap'. An argument is made in favour of Protax in the Discussion, that it provides a probability of group membership, rather than a fixed assignment. However, this argument is spurious; there is no point of calculating this probability if you don't know the groups. The situation may be different at the next stage of barcoding, when incoming sequences are run against a well established system of barcode groups (i.e. once a formal delimitation of the groups has been done), but here it is not useful.

In summary, this is an impressive dataset with potentially exciting results, but the analysis could be done in greater depth and the main conclusions elaborated at great length in the Discussion need more thinking.

Reviewer #2 (Remarks to the Author):

This is a carefully planned and well delivered piece of research – well presented in good and clear language. It is near flawless in presentation and can go direct to acceptance (after a couple of small issues noted in the reattached ms).

I have only additional thought that if author could add a paragraph to discuss biogeography. For instance, do DNA barcodes reveal any biogeographic patterns?

Reviewer #3 (Remarks to the Author):

This is a paper focused on an ambitious undertaking – barcoding and analyzing data for all species in a taxonomic group in Europe. The authors justify the project by stating that understanding patterns of genetic diversity at continental scales for entire faunas is still very rare. The focal group is butterflies, among the best known and most studied insect groups. The message is clear – this effort provides a critical foundation for better understanding of what species illustrate interesting patterns, of what species require further work, and of species of conservation concern, in addition to aiding our general understanding of ecology and evolution. The number of species is so high (459) that details on species serve as examples of general, higher level patterns only possible to describe with this kind of dataset. The authors nicely summarize some key findings about genetic diversity and its faunal patterns across Europe. For example, the association of increased genetic diversity with southern refuges. I think these patterns will be of interest to specialists and non-specialists alike. The writing is clear and analyses sound, and I have only a few relatively minor comments, listed below. However, one larger one involving clarifying the results and interpretation depicted in figure 3 should be addressed.

line 61 species (missing "e")

line 103 – Analysis of rarefaction sampling curves

Figure 3 legend – please give an explanation of trend line and error band. I think more explanation is needed in legend or in text of whether the smoothed curve is being interpreted or the individual points, because they tell slightly different stories (see next comment).

Figure 3 and associated text on lines 122-125 for latitude and lines 149-152.

For latitude, the smoothed line and points shows a strong peak at around 40 degrees latitude, and this is discussed. However, there is a clear second, albeit smaller peak of points and associated small rise in smoothed curve at 55 degrees latitude. This second peak is apparently not discussed.

For longitude, the authors discuss three peaks associated with three peninsulas, and this is seen with the points, although you could also count five separate peaks. The smoothed curve shows two broad peaks fit to the data, but three are discussed.

So in the case of latitude one peak is discussed, but there is evidence of two in the figure, and in the case of longitude three peaks are discussed, but there is evidence of more (points), or fewer (smoothed line), depending on what you focus on.

This should be addressed and text edited as necessary to adjust the interpretation. There is no statistical support reported for optimal number of peaks ($k=10$ is mentioned in methods line 501). The only statistical results about this analysis are Table S4 & S5 (page 504(!) of supplementary materials but no information about optimum number of peaks is indicated.

line 197 p-distance needs to be defined

line 392-393 sister-cryptic species should be re-worded

Use of COI or COI should be consistent throughout manuscript.

line 437-438 – were any samples identified to species mis-identified, and the mis-identification found out through, and fixed in, preliminary analyses?

line 457 and line 109 in supplement: p-distance (- missing)

line 521-522 number should not be wrapped

Table S3 – I think this would be better sorted by taxonomy so it is easier to find taxa of interest (even though they are not all present).

With the supplementary tables being so long I suggest putting the column headers on each page wherever possible.

Reviewers' comments:

Reviewer #1 (Remarks to the Author):

This is an impressive study that takes DNA barcoding to the next level of resolution at the individual genotype, rather than species identification. About a quarter of the 20000+ sequences were newly generated, resulting in a dataset that is largely complete for all species of butterfly across Europe. This allows analyses of patterns of diversity at the levels of mitochondrial haplotypes, which the study exploits nicely to assess latitudinal and longitudinal gradients. This work reveals the expected patterns of high diversity near the southern part of the continent, and some variation longitudinally that were explained by differences in the land area correlated to the Mediterranean peninsulas. We also obtain information on the number of different haplotypes in each species, revealing great differences ranging from a few to several hundred in some cases, and the distribution of common and rare haplotypes. I think it is an excellent resource and a beautiful example of the power of dense mitochondrial barcoding.

Response: Thank you for your comments that help improve the quality of the manuscript. We are glad that you found the work to be overall interesting.

However, the study has some critical flaws that should be addressed. A key conclusion is that mitochondrial variation is not reflecting effective population size. This is supported with a few casual examples, e.g. the fact that fairly widespread species like *Gonepteryx cleopatra* exhibit only a few haplotypes. There is no analysis of any kind beyond these anecdotal observations. Perhaps one could make the opposite argument, i.e. that population size is correlated with effective population size. For example, the widespread *Pyronia cecilia* exhibits 331 haplotypes. But this, again, is just another anecdote. This type of conclusion needs more critical thinking and an analytical framework based in coalescence theory. As it stands, it simply gives ammunition to the idea that DNA barcoding and mitochondrial markers in general are inferior to population studies using nuclear genes and ultimately will restrict its wider use.

Response: We thank you for this excellent observation to which we agree. We have run new analyses testing for the correlation between haplotype richness and species range (used as a proxy for population size). The results show a highly significant correlation even after considering the effect of number of specimens sequenced per species. Thus, we have included a new Table 1 and a new Figure S1a,b. Although, this new result was expected based on theoretical predictions, it is not trivial since many studies failed in finding a clear evidence for correlation between DNA polymorphism and proxies for population size and geographic range. A notable aspect of our result is that it was inferred based on an entire taxonomic group at a continental scale.

Second, the analysis of the taxonomy using PROTAX is a curious approach. The authors aim to test the placement of individual barcode sequences into the named taxa, finding that this works most of the time when these groups are discrete and genetically distant, but not if they are poorly separated. Based on the description of the algorithm in the Methods section, this is not a surprise at all because the method simply gives a probability of the fit in a certain group based on genetic distances. If the Protax placement isn't correct, this is simply the result of the composition of the groups. The problem can only be addressed with a study of what

should be the groups, rather than an algorithm that tests the placement of a sequence into the groups. As the authors are well aware, there are many approaches to address this issue that are widely established in barcoding, including methods based on diagnostic character, tree-based methods, and clustering algorithms such as BIN and ABGD (which are probably closest to what Protax does). Any of these should have been used, but not a method such as Protax that does not address what should be the extent of the groups. As it stands it provides a convoluted analysis of the 'barcoding gap'. An argument is made in favour of Protax in the Discussion, that it provides a probability of group membership, rather than a fixed assignment. However, this argument is spurious; there is no point of calculating this probability if you don't know the groups. The situation may be different at the next stage of barcoding, when incoming sequences are run against a well established system of barcode groups (i.e. once a formal delimitation of the groups has been done), but here it is not useful.

Response: We realized that we did not justify our approach in the introduction. For this reason, we enlarged the introduction and explained why we decided to use PROTAX. Indeed, the taxonomy of a large group such as butterflies over an entire continent cannot be assessed based on a single marker as done by DNA barcoding; the most recent checklist of European butterflies has been assessed based on the consensus among a group of experts (Wiemers et al. 2018) and recognizes species only when they are characterized by divergence in multiple markers.

Moreover, due to several phenomena like introgression, incomplete lineage sorting paraphyletic speciation, the possibility to attribute specimens to a given taxonomic species based on DNA barcoding should not necessarily show a binary response but, in many cases, a continuous probability of attribution could be a more appropriate output. Despite the strong theoretical basis for this phenomenon, the degree to which DNA barcoding attribution has (or not) a binary response has never been assessed over an entire group at the continental scale. This is why we used PROTAX providing unbiased probabilities of correct taxonomic placement based on a priori determined classification (the list of Wiemers et al. 2018).

As a direct comparison, we are now also including an analysis based on the BIN system. We found that a very high number of species in our dataset (36.6%) share BIN with at least another species. This is mostly due to the fact that several species of European butterflies display quite low distances to nearest-neighbor (NN), leading to assignment under the same BIN. However, PROTAX seems able to identify correctly with very high probability many of such cases (as mentioned in the paper, at least 99.9% reliability for distances to NN over 0.61%). We think that a direct comparison between a binary and a probabilistic method allows a novel understanding of the pattern of mtDNA diversification and of the taxonomic value of the identification based on DNA-barcoding data. We are also mentioning that some of the currently recognized species may need further taxonomic study. Thus, the issue of correctly defining species is complex, but DNA barcodes can highlight cases in need of additional study.

In summary, this is an impressive dataset with potentially exciting results, but the analysis could be done in greater depth and the main conclusions elaborated at great length in the Discussion need more thinking.

Reviewer #2 (Remarks to the Author):

This is a carefully planned and well delivered piece of research – well presented in good and

clear language. It is near flawless in presentation and can go direct to acceptance (after a couple of small issues noted in the reattached ms).

Response: We are glad that you found the manuscript interesting and thank you for the comments. We applied all of them and have reformatted the References according to the requirements of Communications Biology. Please find below more detailed responses to some of your comments.

I have only additional thought that if author could add a paragraph to discuss biogeography. For instance, do DNA barcodes reveal any biogeographic patterns?

Response: In this study we have focused on two main topics: i) the possibility to identify specimens using DNA barcoding and the advantages of using a probabilistic approach and on ii) the frequency and distribution of haplotype richness. In this context, the distribution of number of haplotypes across Europe is the biogeographic pattern we studied, which has fundamental implications in biogeography: retreat to southern refugia during Pleistocene and subsequent dispersal of subsets of haplotypes. Comparison among areas in genetic differentiation has been done in a previous paper (<https://doi.org/10.1111/1755-0998.13059>) where, based on a subset of the data analysed here, we were able to retrieve a zoogeographic map for western Europe.

How do you trust identifications of the public COI sequences? There could be some misidentifications.

Response: We agree this is an aspect to carefully consider. The majority of the sequences included in the dataset originate from studies previously published by the authors of the current paper, so we had access to the specimens. For the remaining specimens, we were able to check identifications based on the information (and images) stored in BOLD.

Reviewer #3 (Remarks to the Author):

This is a paper focused on an ambitious undertaking – barcoding and analyzing data for all species in a taxonomic group in Europe. The authors justify the project by stating that understanding patterns of genetic diversity at continental scales for entire faunas is still very rare. The focal group is butterflies, among the best known and most studied insect groups. The message is clear – this effort provides a critical foundation for better understanding of what species illustrate interesting patterns, of what species require further work, and of species of conservation concern, in addition to aiding our general understanding of ecology and evolution. The number of species is so high (459) that details on species serve as examples of general, higher level patterns only possible to describe with this kind of dataset. The authors nicely summarize some key findings about genetic diversity and its faunal patterns across Europe. For example, the association of increased genetic diversity with southern refuges. I think these patterns will be of interest to specialists and non-specialists alike. The writing is clear and analyses sound, and I have only a few relatively minor comments, listed below. However, one larger one involving clarifying the results and interpretation depicted in figure 3 should be addressed.

line 61 species (missing “e”)

Response: Corrected.

line 103 – Analysis of rarefaction sampling curves

Response: Applied.

Figure 3 legend – please give an explanation of trend line and error band. I think more explanation is needed in legend or in text of whether the smoothed curve is being interpreted or the individual points, because they tell slightly different stories (see next comment).

Response: Please see the next comment.

Figure 3 and associated text on lines 122-125 for latitude and lines 149-152.

For latitude, the smoothed line and points shows a strong peak at around 40 degrees latitude, and this is discussed. However, there is a clear second, albeit smaller peak of points and associated small rise in smoothed curve at 55 degrees latitude. This second peak is apparently not discussed.

Response: We revised all this section of the Results and better explained how we obtained the relationship between latitude-longitude and richness and how we generated the graph. Now it is clearer that the graph serves to inspect the significant trend between richness and latitude and longitude. We also updated Fig. 3a in order to provide a more detailed representation of the Pleistocene ice sheets.

“A Smoothed Conditional Means plot using locally estimated scatterplot smoothing (loess, with $\alpha=0.4$) made it possible to inspect the trend of these significant relationships. This analysis showed that haplotype richness was highest (>12 haplotypes per species) at latitudes ranging approximately between 38-47 degrees. This belt included most of the large southern European peninsulas, as well as the Pyrenees, southern Alps, the Balkans, and southern Carpathians. More southerly regions of Europe (<38 degrees) displayed fewer haplotypes and a similar decline was evident at >47 degrees with most regions in Scandinavia displaying less than five haplotypes per species. A smaller peak in haplotype richness around 55 degrees may reflect stochasticity or the presence of a high geomorphological complexity due to several sea straits among mainland areas (Ireland, Britain, Central Europe, Scandinavian Peninsula).”

For longitude, the authors discuss three peaks associated with three peninsulas, and this is seen with the points, although you could also count five separate peaks. The smoothed curve shows two broad peaks fit to the data, but three are discussed.

So in the case of latitude one peak is discussed, but there is evidence of two in the figure, and in the case of longitude three peaks are discussed, but there is evidence of more (points), or fewer (smoothed line), depending on what you focus on.

This should be addressed and text edited as necessary to adjust the interpretation. There is no statistical support reported for optimal number of peaks ($k=10$ is mentioned in methods line 501). The only statistical results about this analysis are Table S4 & S5 (page 504(!) of supplementary materials but no information about optimum number of peaks is indicated.

Response:

We revised the results of this section referring to longitude. Now the peaks discussed agree with those highlighted in the figure. We also clarified that the Smoothed Conditional Means plots of haplotype richness using locally estimated scatterplot

smoothing (with 95% confidence interval) serve to visually inspect the significant trend between richness and latitude-longitude so that in our opinion no complex analysis aimed at detecting the optimal number of peaks is needed. Also because the existence of a latitudinal trend (the main result) is very clear.

line 197 p-distance needs to be defined

Response: p-distance is now explained.

line 392-393 sister-cryptic species should be re-worded

Response: We have reworded that part of the text as it follows:

“Certain sister species or some cryptic species display low minimum interspecific divergence, with some of the best-known cases in European butterflies being.....”

Use of COI or COI should be consistent throughout manuscript.

Response: Applied.

line 437-438 – were any samples identified to species mis-identified, and the mis-identification found out through, and fixed in, preliminary analyses?

Response: Yes, we did find a few such cases that needed correction. Checking the identifications was a long process and we did not keep evidence of all the cases that have been corrected in the newly generated data, but we are now mentioning in the manuscript that some cases had identifications updated following morphological examination.

“Following morphological analyses (especially genitalia morphology), a few specimens turned out to be initially misidentified and this has been corrected in the final dataset (e.g. in genera such as *Plebejus*, *Melitaea* or *Hipparchia*, encompassing several externally very similar species).”

line 457 and line 109 in supplement: p-distance (- missing)

Response: Corrected.

line 521-522 number should not be wrapped

Response: Corrected.

Table S3 – I think this would be better sorted by taxonomy so it is easier to find taxa of interest (even though they are not all present).

Response: Applied.

With the supplementary tables being so long I suggest putting the column headers on each page wherever possible.

Response: We are now providing all large tables in excel format, which should facilitate reader access.

Reviewers' comments:

Reviewer #1 (Remarks to the Author):

The authors made a good effort to respond to the various reviews, in particular to establish the correlation of genetic diversity and population size (using area as a proxy).

I am less satisfied with the response about ProTax. This is still not right (or just poorly explained). The way it is described in the paper ProTax is used for a de novo classification of these groups. In my previous review I pointed out that you can't make a placement of sequences into groups if you don't have some definition of what these groups are beforehand. You need a reference set for that. Presumably Protax can do this to some extent without having a comprehensive reference set by applying a model, but this model needs to be parameterised (a training set of some kind is needed). This doesn't seem to be the case here, and instead all things were grouped based on some unclear probability used in applying this algorithm. This would simply result in some clusters that will fit quite well with the data overall, and not surprisingly the clusters that are more discrete do better than those that aren't.

The comparison with the BIN groups is instructive. This is a method that creates these entities de novo. In this case, the separation of species is less clear than with the Protax analysis. This is in part explained because the BIN analysis doesn't take a probabilistic approach and therefore is not able to 'ignore' cases of haplotype sharing, even if they affect a small minority of haplotypes. But in principle this is the correct approach to take.

If a few sequences don't fit these otherwise well separated groups, this is entirely expected based on the population genetics of speciation. Coalescence of haplotype frequently goes back to a time before the speciation event. This would be dealt with ideally with by a coalescent analysis looking at various parameters of population size, migration, mutation rate etc. but generally you don't have enough data and it's based on only a single marker.

The paper (I think) tries to get around this by using the probabilistic approach of ProTax. [if it had been applied correctly - see above]. But in my view this is creating a reality of these species (a taxon concept for each) that is not justified based on any biological process. It's just a clustering of the haplotypes that are in the mix at this stage, allowing a bit of probabilistic uncertainty about the boundaries based on the frequency of mild outliers. Protax should not be used as a method for species delimitation.

Reviewer #3 (Remarks to the Author):

The authors have addressed my original concerns/suggestions and as I see it, those of the other reviewers.

The introduction is now longer and does require some editing to be more concise. I enjoyed the addition of the northern purity/southern variation predictions.

Lines 74-80, long sentence could be made more clear by breaking it up or reorganizing.

Lines 81-93 "the same reasons" is not concise and I think it could be more clear with a stronger topic sentence for this paragraph

line 87 "was still lacking", is confusing, meaning before this study?

line 84 - reference for theoretical basis?

line 329 - compared to reference 6? Please clarify

line 394 "this approach" should be clarified

Reviewers' comments:

Reviewer #1 (Remarks to the Author):

The authors made a good effort to respond to the various reviews, in particular to establish the correlation of genetic diversity and population size (using area as a proxy).

I am less satisfied with the response about ProTax. This is still not right (or just poorly explained). The way it is described in the paper ProTax is used for a de novo classification of these groups. In my previous review I pointed out that you can't make a placement of sequences into groups if you don't have some definition of what these groups are beforehand. You need a reference set for that. Presumably Protax can do this to some extent without having a comprehensive reference set by applying a model, but this model needs to be parameterised (a training set of some kind is needed). This doesn't seem to be the case here, and instead all things were grouped based on some unclear probability used in applying this algorithm. This would simply result in some clusters that will fit quite well with the data overall, and not surprisingly the clusters that are more discrete do better than those that aren't.

The comparison with the BIN groups is instructive. This is a method that creates these entities de novo. In this case, the separation of species is less clear than with the Protax analysis. This is in part explained because the BIN analysis doesn't take a probabilistic approach and therefore is not able to 'ignore' cases of haplotype sharing, even if they affect a small minority of haplotypes. But in principle this is the correct approach to take.

If a few sequences don't fit these otherwise well separated groups, this is entirely expected based on the population genetics of speciation. Coalescence of haplotype frequently goes back to a time before the speciation event. This would be dealt with ideally with by a coalescent analysis looking at various parameters of population size, migration, mutation rate etc. but generally you don't have enough data and it's based on only a single marker.

The paper (I think) tries to get around this by using the probabilistic approach of ProTax. [if it had been applied correctly - see above]. But in my view this is creating a reality of these species (a taxon concept for each) that is not justified based on any biological process. It's just a clustering of the haplotypes that are in the mix at this stage, allowing a bit of probabilistic uncertainty about the boundaries based on the frequency of mild outliers. Protax should not be used as a method for species delimitation.

Response

Thank you for your comments. We are glad that you liked our approach to the correlation of genetic diversity and population size.

Your concerns relating to the use of PROTAX presume it was employed for the *de novo* classification of groups, but this was not the case. PROTAX is not a species delimitation method. Based on a sequence reference database and an established taxonomy, it estimates the probability with which a sample can be placed in a given taxon. In fact, our established taxonomy followed Wiemers et al. 2018 (reference 22). Therefore, PROTAX was only used to assess identification success using DNA barcodes. The main difference compared to other methods employed to assess identification success is that PROTAX provides a probability of correct identification.

In practice, we tested how reliable is the attribution to the correct species of a given specimen based on the library we assembled. Given the completeness of our library, it is reasonable to assume that the percentages obtained will not change significantly with increasing the sample size and that the values lower than 100% are in fact the effect of biological phenomena (introgression, incomplete lineage sorting, paraphyletic speciation), or of taxonomic uncertainty (e.g. species that should be lumped). Using PROTAX, we were able to show that, in many cases where the identification is not 100%, there is still a high confidence (> 85%) which can be used as a reliable attribution which cannot be obtained by other methods (see lines 388-393 of the track changes manuscript). Furthermore, some species that were placed in the same BIN (due to low interspecific divergence) still had identification probabilities of 100% (lines 395-406 of the track changes manuscript).

Thus, our study only documents the value of PROTAX as a method of assessing identification success, not as delineating species, using our reference library and the taxonomy from Wiemers et al. (2018).

We do appreciate that the subtlety of this distinction was not sufficiently emphasized in our last submission. Accordingly, we revised text in several places (track changes: 91-93, 240242, 375-379, 634-635) to clarify the use of PROTAX. We trust these changes address your concern.

Reviewer #3 (Remarks to the Author):

The authors have addressed my original concerns/suggestions and as I see it, those of the other reviewers.

The introduction is now longer and does require some editing to be more concise. I enjoyed the addition of the northern purity/southern variation predictions.

Response: Thank you for your comments. The Introduction is now more concise.

Lines 74-80, long sentence could be made more clear by breaking it up or reorganizing.

Response: This sentence was too long; we revised the text and split it into two sentences (lines 71-78 of the revised manuscript with track changes).

Lines 81-93 "the same reasons" is not concise and I think it could be more clear with a stronger topic sentence for this paragraph

Response: We are now writing: "The possibility to assign specimens to a species based on its DNA barcode is not always a binary response but can be better viewed as a continuous probability of correct attribution." (lines 82-85 of the revised manuscript with track changes).

line 87 "was still lacking", is confusing, meaning before this study?

Response: We deleted this sentence to shorten the Introduction.

line 84 - reference for theoretical basis?

Response: We rephrased this part (lines 82-88 of the revised manuscript with track changes).

line 329 - compared to reference 6? Please clarify

Response: We have added "compared to reference⁶". We made the same revision two lines above (lines 321-322 of the revised manuscript with track changes).

line 394 "this approach" should be clarified

Response: We agree and rewrote this part as follows:

"As a probabilistic approach, PROTAX is based on a given (predefined) taxonomy and a set of reference sequences, and its accuracy increases when taxonomic knowledge and coverage, as well as sampling quality, are high⁴⁸. Therefore, our DNA barcode library of European butterflies is well suited for the application of this method."

(lines 375-379 of the revised manuscript with track changes).

Reviewers' comments:

Reviewer #1 (Remarks to the Author):

I appreciate the recent additions to explain the use of PROTAX. This is much clearer now. So, an existing species delimitation is used, each species identified according to this existing taxon concept, and each barcode is associated with a species in this existing system. Correct so far?

The probabilistic approach comes in by using PROTAX that assigns each sequence to a hierarchical classification system. As each barcode is assigned to a given species, the 'training set' is highly resolved, i.e. all sequences are assigned to a species. In the next step, each of the 24000 sequences is taken in turn and tested against these highly resolved groupings. For the most part, the sequences fit exactly in one of these clusters, as one would expect, except in a few cases where a sequence is quite divergent from any others and doesn't fit in the existing groups. This apparently never happens in cases of divergences of 0.61%.

That's an interesting result. It says that any new sequence of that divergence or less probably fits neatly into this barcoding system.

It is less clear what this says about the classification system itself. The classification system used is directly designed in this study, by linking the morphological classification to the barcodes generated here (as each barcode sequence is assigned to one of these morphologically defined species). This means, the morphology is accepted as true and the barcodes are grouped in a way that match the morphology.

The probabilistic placement of sequences is only with respect to this set, not the groups that might be obtained from the barcodes themselves..

Hence, the comparison with the BIN system is misleading. The BIN groups are used for comparing the de novo generated barcode groups against the morphological system. So here the two character systems (morphology and mitogenomes) "are allowed to clash", and they do. The differences in the results of both approaches have nothing to do with using a probabilistic versus "absolute" approach, as claimed in the paper. The sentence used to explain the differences found with the BIN method ("These cases of BIN sharing despite diagnostic divergence are not a surprise as the BIN algorithm was designed to take a conservative approach to species estimation." (line 405) is not correct. The reason is that this method is not grouping the barcodes according to the morphologically identified species, unlike the approach used to generate the groups that are tested with the PROTAX assignment.

[Possibly I am still misunderstanding what was done to get the reference set but this is my best guess based on the sentence "We used all 22,074 DNA barcodes in our library as reference sequences." (Line 644)]

Reviewers' comments:

Reviewer #1 (Remarks to the Author):

I appreciate the recent additions to explain the use of PROTAX. This is much clearer now. So, an existing species delimitation is used, each species identified according to this existing taxon concept, and each barcode is associated with a species in this existing system. Correct so far?

Response: With regards to the use of PROTAX, we are glad that the new version of the manuscript is clearer. Indeed, we are using an existing species delimitation – i.e. each barcode is assigned to a species which corresponds to the most up to date checklist published by Wiemers et al. (2018). For further information regarding the way we assign specimens to individual species please see below.

The probabilistic approach comes in by using PROTAX that assigns each sequence to a hierarchical classification system. As each barcode is assigned to a given species, the 'training set' is highly resolved, i.e. all sequences are assigned to a species. In the next step, each of the 24000 sequences is taken in turn and tested against these highly resolved groupings.

Response: The interpretation is correct, but please note that the training set is not necessarily resolved in all cases, but in the most sensible way based on current knowledge. Indeed, we identified specimens based on a hierarchical approach using morphology, life history, distribution, nuclear and mitochondrial DNA (in this order) to attribute specimens to a given species in line with the checklist published by Wiemers et al. (2018). In practice, when information regarding external and/or internal (genitalia) morphology is sufficient to identify a specimen, we attributed the specimen to a species. To accomplish this, genitalia of 1,490 specimens were examined. In problematical instances (e.g. cryptic species), information relating to species ecology and life history (e.g. larval food plant, habitat, phenology) was used; in addition we compared the collection site with the known distribution of species; we also deliberated with information collected from nuclear DNA (when available). Mitochondrial DNA was used as a last resort (always in combination with other data) in only a small number of specimens, including taxa for which previous studies have shown a virtually perfect correlation between DNA barcodes and species delimitation based on other criteria (e.g. *Leptidea sinapis*, *L. reali*, *L. juvernica*). Using this hierarchical procedure, we can identify cases of introgression (morphology of species A and COI of species B). In these cases, PROTAX generates a tree, showing a clade of several specimens identified as B and one identified as A. Consequently, a sequence belonging to species B cannot be allocated to a named species with 100% certainty because of the presence of species A in its clade. The revised manuscript now explains in more detail how we performed the identification of specimens in our barcode library, both at the beginning of the Results and in the Methods.

For the most part, the sequences fit exactly in one of these clusters, as one would expect, except in a few cases where a sequence is quite divergent from any others and doesn't fit in

the existing groups. This apparently never happens in cases of divergences of 0.61%.

That's an interesting result. It says that any new sequence of that divergence or less probably fits neatly into this barcoding system.

Response: 0.61% refers to minimum divergence to the nearest specimen of another species in our dataset. This value (0.61%) defines the level of sequence differentiation between the species in our dataset that allowed a certain determination (100% success) by PROTAX. By comparison, species with a lower divergence (<0.61%) showed reduced resolution (potentially due to incomplete lineage sorting, introgression, bad taxonomy etc).

It is less clear what this says about the classification system itself. The classification system used is directly designed in this study, by linking the morphological classification to the barcodes generated here (as each barcode sequence is assigned to one of these morphologically defined species). This means, the morphology is accepted as true and the barcodes are grouped in a way that match the morphology.

Response: Thank you for this comment. We realized that the description of our hierarchical procedure was fundamental to better understanding the PROTAX results (see above) and we added a paragraph in the methodology describing this (lines 509-522 of the revised manuscript-track changes).

The probabilistic placement of sequences is only with respect to this set, not the groups that might be obtained from the barcodes themselves.

Response: Indeed, the probabilistic placement of sequences with the correct species is only with respect to our dataset. Consequently, the results will depend on the completeness and correctness of the dataset. In this respect, we have done our best to assign the barcodes to the correct species based on the criteria discussed above. The new version of the manuscript stresses it more:

“As a probabilistic approach, PROTAX is based on a given (predefined) taxonomy and a set of reference sequences, and its accuracy increases when taxonomic knowledge and coverage, as well as sampling quality, are high⁴⁸. Therefore, our comprehensive DNA barcode library of European butterflies coupled with PROTAX should represent a powerful method of attributing specimens to their correct species.”

Hence, the comparison with the BIN system is misleading. The BIN groups are used for comparing the de novo generated barcode groups against the morphological system. So here the two character systems (morphology and mitogenomes) "are allowed to clash", and they do. The differences in the results of both approaches have nothing to do with using a probabilistic versus "absolute" approach, as claimed in the paper. The sentence used to explain the differences found with the BIN method ("These cases of BIN sharing despite diagnostic divergence are not a surprise as the BIN algorithm was designed to take a

conservative approach to species estimation." (line 405) is not correct. The reason is that this method is not grouping the barcodes according to the morphologically identified species, unlike the approach used to generate the groups that are tested with the PROTAX assignment.

Response: We agree and removed the sentence from line 405. We still retain the BIN analysis, as we believe that your earlier suggestion to add this analysis was a valuable addition to the manuscript.

[Possibly I am still misunderstanding what was done to get the reference set but this is my best guess based on the sentence "We used all 22,074 DNA barcodes in our library as reference sequences." (Line 644)]

Response: You are correct, the reference set was our DNA barcode library. To make this point clearer, we now state: "We used all 22,074 DNA barcodes in our library as a reference set". Moreover, as explained above, in this new version of the manuscript the construction of the reference library is described in greater detail.

REVIEWERS' COMMENTS:

Reviewer #1 (Remarks to the Author):

Based on these responses, my interpretation of how the PROTAX methodology was applied seems accurate. We have a dataset of sequences that each has been assigned to a species and the PROTAX probability of assignment is performed against this dataset. If the species assignment was based on the COI sequences this approach is circular and the authors now seem to agree with this.

The new version of the paper emphasises that the species assignment was not actually based on the COI sequences used for the PROTAX assignment. Instead, a combination of morphological, ecological, geographical and nuclear DNA information is used to do the assignment, although where this information is not producing the required information, COI sequences are used, after all. We are told that a small number of cases are affected, although these may in fact be the critical ones, retaining some of the circularity. [A rather confusing addition in this version is the term "hierarchical approach" for this procedure of defining the species, which of course differs from the use of the term "hierarchical" in the way PROTAX conducts taxonomic assignment.]

This analysis is set against the BIN assignment, i.e. the de novo generation of haplotype groups based on a graph theory approach. I do agree this analysis is useful, but its interpretation needs a comment, which is now missing. The suggestion implied in the paper is that PROTAX can deal with any (small) inter-specific distance, so long as the sequences are assigned to the correct species – i.e. assigned based on the "hierarchical" species assignment used by the authors. Whereas a procedure that is uniformly applied to all data like BIN, without any human intervention, will get it wrong if the entities are not of uniform composition, especially if some interspecific distances are very small. The statement that 63.4% of species are affected is important: a uniform treatment of the data obviously is inappropriate, not just based on these findings but given our knowledge of the coalescent process underlying the species evolution.

The manual/hierarchical assignment of haplotypes to a species can alleviate this, if we "know" what the species are. This assignment of haplotypes using external information can even include cases of haplotypes that are identical to two or more species, or even cases of non-monophyly of haplotypes corresponding to a single species. These cases would presumably be the ones that PROTAX would, respectively, assign with lower probability or would assign incorrectly. These cases are to be expected due to the evolutionary processes of delayed lineage sorting etc. and, just like the BIN procedure, PROTAX can't deal with them due to the hierarchical assignment of the algorithm.

In conclusion, it is required for the final version of the paper to add a paragraph about the use of PROTAX and the need for an independent haplotype assignment when using this approach. Please be clear where the method can alleviate problems arising with the BIN approach (and other de novo methods), and where it can't, and present the reasons. It is not correct that PROTAX can deal with small distances per se, as implied in line 378, but instead this is a consequence of the manual species assignment in the reference set that can separate species no matter what is their distance. Please also make sure the reader understands this thinking: in the current description of the hierarchical approach of species assignment (line 509) the rationale would not be clear to the reader without more emphasis. Finally, the paper should be more precise about what is the advantage of PROTAX. In the Conclusion it is said that "Use of PROTAX established that this library is highly effective (>95%) in identifying European butterflies." I think it's the other way around; the library is highly effective/informative when used with PROTAX, at least with the current dataset. Obviously if new haplotypes are included (or existing ones are found in different species), the method may have a greater chance to fail when trained on the current reference set.

REVIEWERS' COMMENTS:

Reviewer #1 (Remarks to the Author):

Response: We thank you for the positive evaluation of the latest version of our manuscript. We are also glad that we were eventually able to explain how the PROTAX approach can represent a useful interface between an established taxonomy and DNA barcoding.

Based on these responses, my interpretation of how the PROTAX methodology was applied seems accurate. We have a dataset of sequences that each has been assigned to a species and the PROTAX probability of assignment is performed against this dataset. If the species assignment was based on the COI sequences this approach is circular and the authors now seem to agree with this.

Response: In the Introduction we are mentioning that: “To make this assessment possible, PROTAX²³ is a method that gives probabilistic taxonomic assignment (i.e. identification) for a sequence based on a given taxonomy and a set of reference sequences that were identified a priori.”

Moreover, in the revised manuscript at line 109 (revised manuscript, track changes), we changed “DNA barcodes” with “the current DNA barcode library” since, indeed, it is not only the sequences that allow the identification, but also the taxonomic assignment of specimens based on other markers.

The new version of the paper emphasises that the species assignment was not actually based on the COI sequences used for the PROTAX assignment. Instead, a combination of morphological, ecological, geographical and nuclear DNA information is used to do the assignment, although where this information is not producing the required information, COI sequences are used, after all. We are told that a small number of cases are affected, although these may in fact be the critical ones, retaining some of the circularity. [A rather confusing addition in this version is the term “hierarchical approach” for this procedure of defining the species, which of course differs from the use of the term “hierarchical” in the way PROTAX conducts taxonomic assignment.]

Response: We agree and replaced the term “hierarchical approach” with “integrative approach”. In order to avoid confusion, we reserved the word “hierarchical” for the way PROTAX conducts taxonomic assignment.

This analysis is set against the BIN assignment, i.e. the de novo generation of haplotype groups based on a graph theory approach. I do agree this analysis is useful, but its interpretation needs a comment, which is now missing. The suggestion implied in the paper is that PROTAX can deal with any (small) inter-specific distance, so long as the sequences are assigned to the correct species – i.e. assigned based on the “hierarchical” species assignment used by the authors. Whereas a procedure that is uniformly applied to all data like BIN, without any human intervention, will get it wrong if the entities are not of uniform

composition, especially if some interspecific distances are very small. The statement that 63.4% of species are affected is important: a uniform treatment of the data obviously is inappropriate, not just based on these findings but given our knowledge of the coalescent process underlying the species evolution.

The manual/hierarchical assignment of haplotypes to a species can alleviate this, if we “know” what the species are. This assignment of haplotypes using external information can even include cases of haplotypes that are identical to two or more species, or even cases of non-monophyly of haplotypes corresponding to a single species. These cases would presumably be the ones that PROTAX would, respectively, assign with lower probability or would assign incorrectly. These cases are to be expected due to the evolutionary processes of delayed lineage sorting etc. and, just like the BIN procedure, PROTAX can’t deal with them due to the hierarchical assignment of the algorithm.

In conclusion, it is required for the final version of the paper to add a paragraph about the use of PROTAX and the need for an independent haplotype assignment when using this approach. Please be clear where the method can alleviate problems arising with the BIN approach (and other de novo methods), and where it can’t, and present the reasons. It is not correct that PROTAX can deal with small distances per se, as implied in line 378, but instead this is a consequence of the manual species assignment in the reference set that can separate species no matter what is their distance.

Please also make sure the reader understands this thinking: in the current description of the hierarchical approach of species assignment (line 509) the rationale would not be clear to the reader without more emphasis. Finally, the paper should be more precise about what is the advantage of PROTAX.

Response: We agree and we inserted new text addressing these comments at lines 323-342, 363-391 and 523-524 (revised manuscript, track changes).

We think that now both the previous assumptions and the advantages of using PROTAX are clarified.

In the Conclusion it is said that “Use of PROTAX established that this library is highly effective (>95%) in identifying European butterflies.” I think it’s the other way around; the library is highly effective/informative when used with PROTAX, at least with the current dataset. Obviously if new haplotypes are included (or existing ones are found in different species), the method may have a greater chance to fail when trained on the current reference set.

Response: We agree and changed to: “When used with PROTAX, this library is highly effective (>95%) in assigning unknown *COI* sequences to their correct species, as defined according to current taxonomy.”